# Transcription-coupled repair and mismatch repair contribute towards preserving genome integrity at mononucleotide repeat tracts

Ilias Georgakopoulos-Soares[1,2], Gene Koh [1,3,4], Sophie E. Momen[3,4], Josef Jiricny[5], Martin Hemberg [1✉] & Serena Nik-Zainal [3,4✉]

The mechanisms that underpin how insertions or deletions (indels) become fixed in DNA have primarily been ascribed to replication-related and/or double-strand break (DSB)-related processes. Here, we introduce a method to evaluate indels, orientating them relative to gene transcription. In so doing, we reveal a number of surprising findings: First, there is a transcriptional strand asymmetry in the distribution of mononucleotide repeat tracts in the reference human genome. Second, there is a strong transcriptional strand asymmetry of indels across 2,575 whole genome sequenced human cancers. We suggest that this is due to the activity of transcription-coupled nucleotide excision repair (TC-NER). Furthermore, TC-NER interacts with mismatch repair (MMR) under physiological conditions to produce strand bias. Finally, we show how insertions and deletions differ in their dependencies on these repair pathways. Our analytical approach reveals insights into the contribution of DNA repair towards indel mutagenesis in human cells.

[1] Wellcome Sanger Institute, Wellcome Genome Campus, Hinxton CB10 1SA, UK. [2] Department of Bioengineering and Therapeutic Sciences, University of California San Francisco, San Francisco, CA 94158, USA. [3] Academic Department of Medical Genetics, The Clinical School, University of Cambridge, Cambridge CB2 0QQ, UK. [4] MRC Cancer Unit, The Clinical School, University of Cambridge, Cambridge CB2 0XZ, UK. [5] Institute of Molecular Life Sciences, University of Zurich and Institute of Biochemistry, ETH Zurich, CH-8093 Zurich, Switzerland. ✉email: mh26@sanger.ac.uk; sn206@cam.ac.uk

Mutations are not randomly distributed across the cancer genome. Their distribution is influenced by genomic, epigenomic and cellular physiological factors such as replication and transcription[1–4]. Transcription has been implicated in contributing to mutational strand asymmetries reflecting biases in DNA damage (transcription-associated damage) and DNA repair mechanisms (transcription-coupled repair) between the two strands[3–5].

In this area, while substitutions in human cancers have been extensively studied, insertions/deletions (indels) have remained comparatively under-explored. This was historically due to the relative difficulty in obtaining high-quality indel data, further restricted by a limited repertoire of approaches to analyse indels as extensively as substitutions. Nevertheless, indels are common in human cancers and their location and sequence composition are non-random. Thus, like substitutions, they provide important insights into the mutational processes that have shaped the landscape of cancer genomes.

Here, we demonstrate that there is transcriptional strand asymmetry in the distribution of mononucleotide repeat tracts within the reference genome. We also observe transcriptional strand asymmetry in insertions and deletions at mononucleotide repeat tracts across cancer types, and are able to attribute the relative contributions of transcription-coupled nucleotide excision repair (TC-NER) and mismatch repair (MMR) pathways to indel patterns in human somatic cells.

## Results

### Landscape of insertions and deletions across human cancers.
We utilised 2,416,257 indels from a highly curated set of 2575 whole-genome-sequenced (WGS) cancers of 21 different cancer types. Median indel number per tumour was 386, corresponding to a conservative indel density of 0.127 per Mb per cancer genome. Deletions (median 222) were more prevalent than insertions (median 124) in the majority of cancers (Mann–Whitney U $p$ value < 0.05, Fig. 1a and Supplementary Fig. 1a). Moreover, deletion size showed greater variability than insertion size across and within tumour types (Fig. 1b and Supplementary Fig. 1b, c, Levene's test, $p$ value < 0.05).

This first observation can be broadly explained by already-known mechanisms that generate indel lesions. Replication-related DNA polymerase slippage errors running through microsatellites tend to cause deletions[6], because single-stranded DNA ahead of a polymerase can twist, causing a single repeat unit of a run of mono- or dinucleotides in the template strand to loop out. A polymerase passing over such a loop would generate a deletion[6–9]. Because these small insertion/deletion loops (IDLs) are efficiently repaired by MMR, the density of deletions in microsatellites is higher in cells lacking MMR. This phenomenon is referred to as microsatellite instability (MSI). The likelihood of formation of such loops increases with the length of the repeat and we confirm this by showing that indel frequency is augmented with increasing lengths of polynucleotide repeat tracts (Supplementary Fig. 1d, e) and is more pronounced in MMR-deficient samples. By contrast, double-strand breaks (DSB) can give rise to deletions if repaired by non-homologous end-joining, or if addressed by homology-directed sub-pathways such as single-strand annealing or micro-homology-directed end-joining[1,10–15]. The latter result in larger deletions (3 bp in size or more), thus explaining the broader spectrum of observed deletion sizes (Fig. 1b, Levene's test, $p$ value < 0.05).

In general, it is more difficult to create an insertion. Transient dissociation of the primer and template strands and reannealing of the primer in a wrong register within the microsatellite could cause both an insertion or a deletion. These are less likely to arise during normal replication[16] because the end of the primer strand is tightly bound by the replisome. Thus, the occurrence and the relative frequencies of indels and their size spectra can be explained by known mechanisms.

In addition to classical contributions of replication and DSB-repair pathways to indel formation, we introduce another dimension to exploratory analyses of indel mutagenesis: the contribution of transcription. Transcription has been implicated in asymmetric distribution of substitutions between strands for decades[4,17,18]. In particular, TC-NER is believed to preferentially repair DNA damage on the template (transcribed or non-coding) strand. TC-NER activity is thus inferred from the excess of mutagenesis on the non-template (coding) as compared with the template (non-coding) strand, particularly for those environmental mutagens where the target of primary DNA adduct formation is known. For example, guanines adducted by tobacco carcinogens result in an excess of G > T mutations on the non-template strand[19–21]. Likewise, primary covalent modifications of

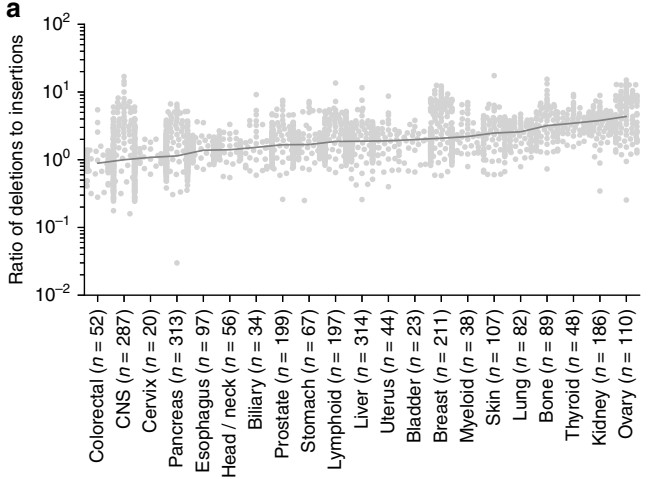

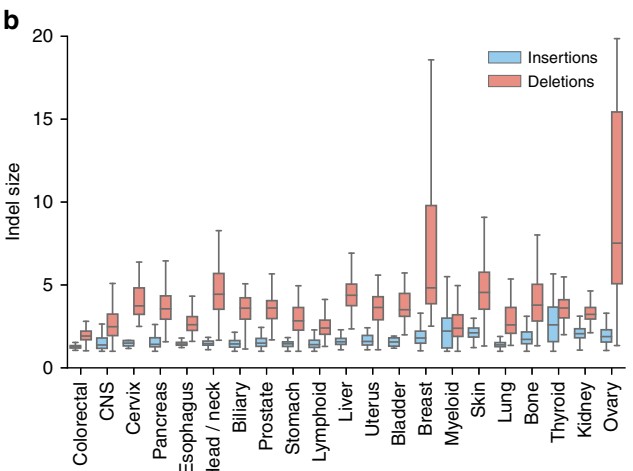

**Fig. 1 Indel characteristics across cancer types. a** The ratio of deletions to insertions for each tumour type. (Mann–Whitney U test, $p$ value < 0.05 per cancer type). **b** Distribution of the size of insertions and of deletions for each tumour type. Deletions displayed greater size variance in comparison to insertions across cancer types (Levene's test, $p$ value < 0.05) and for individual cancer types ($p$ value < 0.001 in breast, pancreas, liver, ovary, skin, lung, cervix, bone, head/neck, colorectal, $p$ value < 0.05 in biliary and lymphoid cancers).

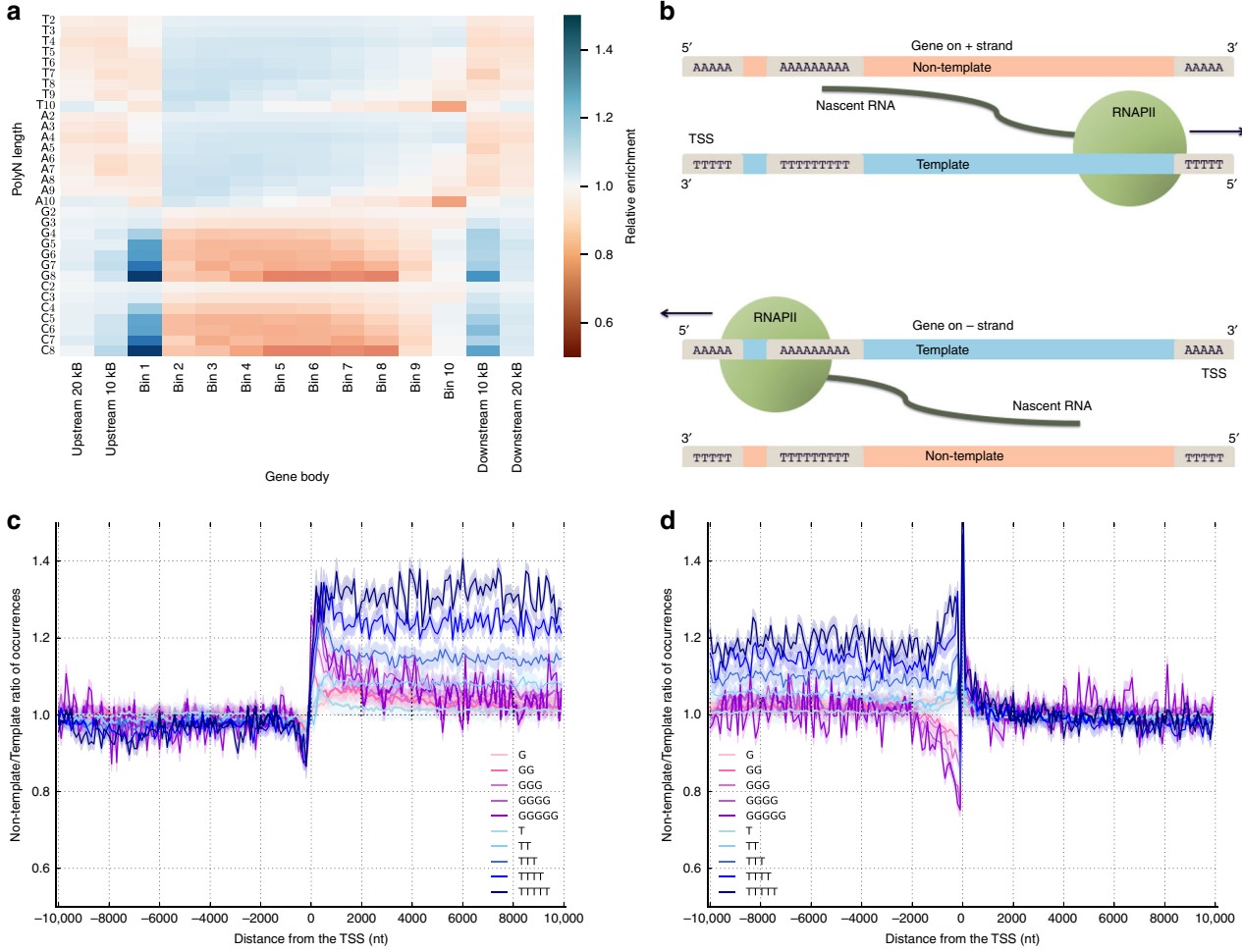

**Fig. 2 Strand asymmetries of polynucleotide (polyN) repeat tracts within transcribed regions. a** Enrichment of various polyN motifs across genes. Each gene is divided into ten bins, and two additional bins are added at either end of each gene. For any given bin, blue indicates relative enrichment in comparison to all other bins for that polyN, whereas red indicates relative depletion. **b** Scheme depicting the identification of polyN motifs on the template (blue) or non-template (orange) strands, dependent on the direction of the gene. RNA-polymerase II (RNAPII) binds to the template strand and mediates transcription. Thus, in the panel above, where the gene is on the (+) strand, the polyA tracts are on the non-template strand. In the panel below, where the gene is on the (−) strand, the polyA tracts are on the template strand. **c** Density of polyT and polyG motifs around the transcription start site (TSS). The gradient of pink to purple represents polyG tracts of 1–5 bp length, whereas the gradient of light blue to dark blue represents polyT tracts of 1–5 bp length. Error bars represent standard error from 1000-fold bootstrapping. **d** Density of polyT and polyG motifs around the transcription end site (TES). Error bars represent standard error from 1000-fold bootstrapping.

cytosines forming 6,4 pyrimidine–pyrimidone dimers and cyclobutane pyrimidine dimers by ultraviolet light are preferentially repaired on the template strand resulting in an excess of C > T transitions on the non-template strand[22]. However, to the best of our knowledge, transcriptional strand asymmetry in indels has not been investigated, primarily due to the technical challenge of being unable to orientate each indel with respect to transcriptional strand.

**Asymmetries of repetitive tracts in the reference genome**. We set out to determine transcriptional strand asymmetry of indels by focusing on mononucleotide repeats of up to ten base pairs in length. We first analysed the distribution of mononucleotide repeats across the gene body in the reference human genome. Each gene was divided into ten equal-sized bins to correct for differences in gene length. Two additional bins were added upstream of the transcription start site (TSS) and two downstream of the transcription end site (TES), each 10 kB in length, resulting in a total of 14 bins.

We observed a strong enrichment of polyG/polyC motifs directly upstream and downstream of the TSS and downstream of the TES; this contrasted with the distribution of polyT/polyA motifs, which were found to be enriched throughout gene body (Fig. 2a and Supplementary Fig. 3a).

We calculated the frequencies of each polyN motif (where N is any nucleotide) on the template and non-template strands in the reference genome. Because the direction of transcription for each gene in the genome is known, each polyN motif can be orientated (Fig. 2b). For example, for a gene on the (+) strand, the template strand is the (−) strand. A polyT motif that is on the (−) strand of this gene is therefore described as being on the template strand. It can also be described as a polyA motif on the non-template strand (Fig. 2b). Using this reasoning, we assigned each polyN motif to either the template or non-template strand of the reference genome. If there were no asymmetries, polyN tracts would occur with equal probabilities on either strand.

Intriguingly, we found that polyT motifs displayed a bias towards the non-template strand in the reference genome, with a non-template to template asymmetry enrichment for short

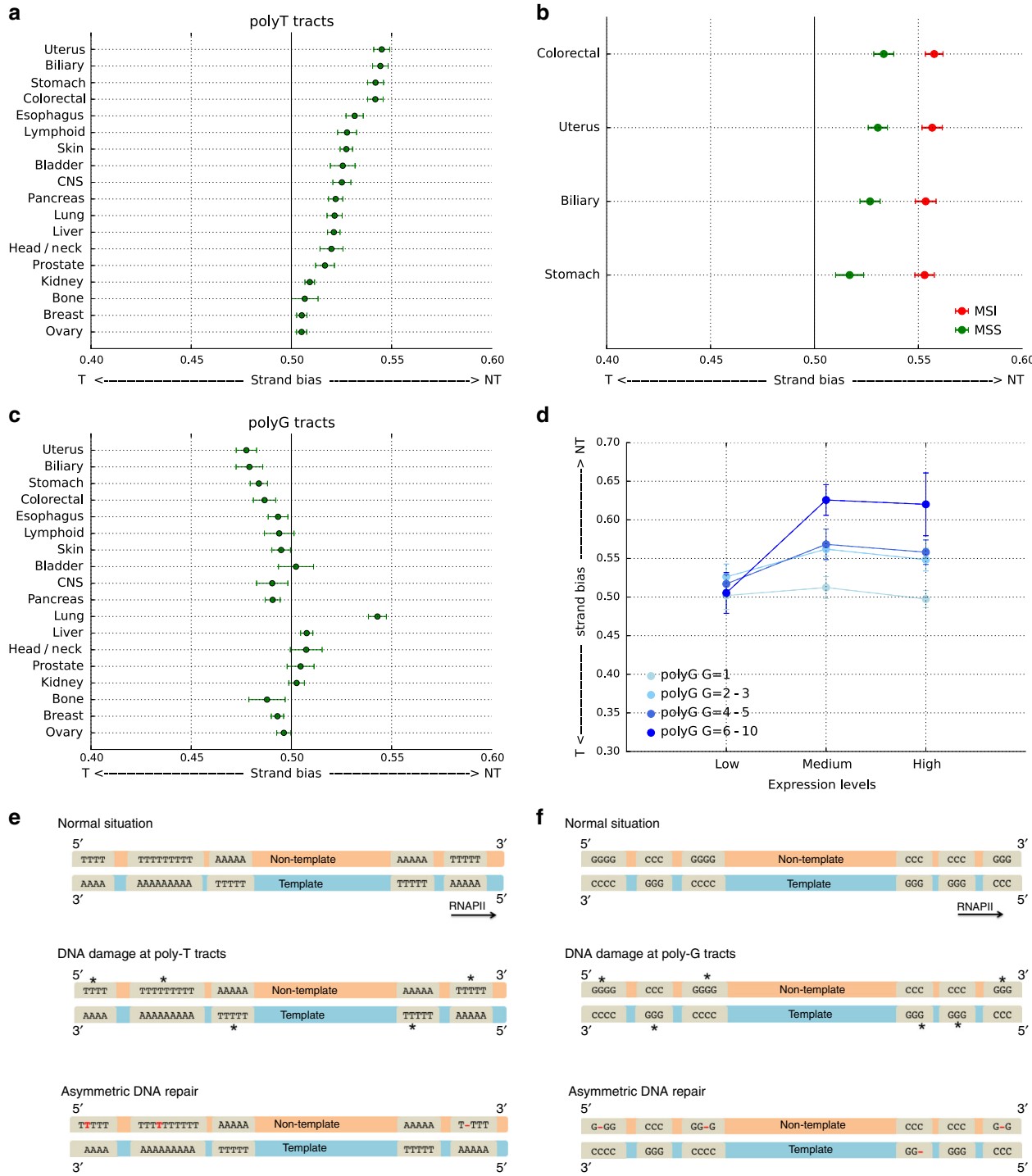

polyT motifs of ~1.15-fold (Fig. 2c, d). This was tract-length-dependent, where longer repetitive tracts were associated with greater strand bias of up to ~1.4-fold at >5nt polyT motifs (weighted average asymmetry of 1.14-fold, Fig. 2b and Supplementary Fig. 3b–g). In contrast, we did not observe a similarly pronounced asymmetry of polyG motifs across gene bodies, although a skew in polyG motifs was noted at the boundaries of gene bodies, in keeping with previous reports of GC-skewing at either end of genes[23] (Fig. 2c, d and Supplementary Figs. 2a, b, 3d–g). The marked variation in strand distributions of the polyN motifs in the reference genome is appreciated particularly around the TSS and TES (Fig. 2c, d and Supplementary Fig. 3d–g).

**Transcriptional strand asymmetries of small indels occur at polynucleotide repeat tracts.** We next investigated whether there was strand asymmetry for indel occurrences at polyN tracts. All analyses henceforth, correct for the skewed background distributions of polyT and polyG motifs. Across cancers, polyT motifs of 2–10 bp in length were consistently more mutable on the non-template strand (binomial test, $p$ value $< e-5$). Strand asymmetry was more pronounced for longer polyT tracts in all cancers (Kruskal–Wallis $H$-test with Bonferroni correction, $p$ value $< e-9$) (Supplementary Fig. 4a). The levels of asymmetry varied by cancer type, with increased indel mutagenesis on the non-template over the template strand ranging from 2.1% in ovarian cancers to 16.5% in uterine cancers (Fig. 3a, e). This was

**Fig. 3 Transcriptional strand asymmetry of indels that occur at polyN tracts across multiple cancer types. a** Transcriptional strand asymmetries of indels occurring at polyT motifs. Average bias is shown with error bars showing standard deviation after 1000 bootstraps. Myeloid, cervix and thyroid cancers were excluded due to low numbers of total indels (Supplementary Table 1). T template, NT non-template. Strand bias was calculated as mutational density of non-template strand over total mutational density (of non-template and template strands). **b** Strand bias of MSI and MSS samples in stomach, biliary, uterus and colorectal tumours (Mann–Whitney U p value < 0.001 in all cases, Bonferroni corrected). **c** Transcriptional strand asymmetries of indels occurring at polyG motifs. Average bias is shown, with error bars showing standard deviation from bootstrapping. **d** Relationship between indel strand bias and gene expression levels in lung cancer (Mann–Whitney U p value < 0.001 for comparisons between low and medium expressed genes and between medium and highly expressed genes) according to length of polyG tracts (Kruskal–Wallis H-test with Bonferroni correction, p value < 0.001 for medium and high expression genes, p value > 0.05 for low expression genes). **e** Scheme depicting mechanism of indel mutagenesis at polyT tracts. DNA damage, shown as asterisks (*) that arise at T nucleotides of polyT tracts can occur on both template and non-template strands. The subsequent DNA repair, postulated to be TC-NER, results in preferential correction of DNA damage on the template strand, leaving T insertions (highlighted in as red T's) and T deletions (shown as red −) on the non-template strand. **f** Schematic depicting mechanism of indel mutagenesis at polyG tracts in lung cancers from smokers. DNA damage in the form of adducted guanines (*) is asymmetrically repaired by TC-NER, with preferential repair of the template strand, thus accumulating more G indels on the non-template strand.

surprising, given that the prevailing dogma on indel formation, particularly at polynucleotide repeat tracts, involves the formation of small IDLs that are substrates for MMR[24–27]. Rather, our analysis showing marked transcriptional strand asymmetry implicates either the activity of TC-NER at these motifs or the activity of transcription-associated damage.

We noted that uterine, colorectal, biliary and stomach cancers showed the highest levels of transcriptional strand asymmetry (binomial test with Bonferroni correction, p value < 0.001 for all four cancer types), with 16.5, 15.5, 16.3 and 15.5% more indels occurring on non-template than template polyT motifs (Fig. 3a). Notably, these cancer types are often associated with incidences of MMR deficiency[28]. To explore the contribution of MMR to transcriptional strand asymmetries of indels at polyT tracts, we compared samples with MMR deficiency (or MSI) with microsatellite stable (MSS) samples. Surprisingly, in MSI samples, transcriptional strand bias towards the non-template strand for polyT motifs was more pronounced than in MSS samples, with a 7.9–12.8% increased indel occurrence (Fig. 3b), (Mann–Whitney U p value < 0.001 in all cases, Bonferroni corrected). This suggests that not only is TC-NER implicated in the repair at polyN motifs, it is also dependent on the normal physiological functioning of MMR. In the absence of MMR, damage at these sites relies more heavily on TC-NER alone, resulting in an increase in strand bias.

**Nucleotide excision repair and MMR influence the indel landscape.** To validate this hypothesis regarding the reliance of TC-NER on MMR, we examined experimentally-generated mutation patterns from CRISPR-Cas9 knockouts of a human cancer cell line, HAP1 (ref. [29]). We would expect the presence of transcriptional strand bias under normal conditions but that the magnitude of the effect would be increased in MMR gene knockouts. Indeed, our analytical findings are recapitulated in the experimental setting. In a knock-out model of MutS homolog 6 (*MSH6*), a key MMR gene, 1663 indels occurred on the non-template strand polyT tracts, whereas 1165 indels occurred on template polyT tracts, corresponding to a 16.9% corrected increase in frequency on the non-template strand (binomial test, p value < e − 6), a similar magnitude to that observed in cancers. However, when this is divided by polynucleotide tract lengths (T, TT, TTT, $T_n$), the numbers are low and the experimental data is under-powered to demonstrate the effect at all repeat lengths, even though the effect is there in aggregate.

Interestingly, in contrast to polyT motifs, we did not observe transcriptional strand bias for indels at polyG motifs across cancers (binomial test, p value > 0.05) (Fig. 3a, c, binomial test, p value < e − 5) (Supplementary Fig. 4b), with lung cancers being the notable exception; they exhibited a large excess of G indels on the non-template strand (15.8% greater indel occurrence at

polyGs on non-template compared with template strand) (binomial test with Bonferroni correction, p value < e − 30) (Fig. 3c). This pattern of asymmetry mirrors what was observed for G > T substitutions in lung cancers, which are attributed to the formation of bulky adducts on guanines from tobacco-related carcinogens. This type of helix-distorting damage is classically repaired by TC-NER[19–21]. The observation of transcriptional strand asymmetry for G indels at polyG tracts in tobacco-associated lung cancers reinforces how TC-NER can be involved in maintenance of genome integrity at polyN motifs, and could therefore also be acting at polyT tracts as hypothesised earlier.

To validate this observation of indel transcriptional strand asymmetry with tobacco exposure, we analysed indel mutational profiles of non-cancerous human cells exposed to various polycyclic aromatic hydrocarbons (PAHs) including benzo[a]pyrene [0.39 and 2 μM] and benzo[a]pyrene diolepoxide [0.125 μM], believed to be the carcinogenic components of tobacco smoke. We observed that 77 indels occurred on non-template polyG tracts, in contrast to only 39 indels on template polyG tracts. This corresponds to nearly double the number of indels on the non-template strand over the template strand (binomial test, p value < 0.001), supporting our analytical observations of in vivo patterns derived from studying human cancers (Fig. 3c, f).

The activity of TC-NER is linked to gene expression levels[30] where higher levels of transcription are associated with increased TC-NER activity. To seek further support that TC-NER plays a role in the repair of polyG motifs in lung cancers, we explored the degree of asymmetry in relation to gene expression levels. We used gene expression data from a representative cell-of-origin (Supplementary Table 2). In keeping with our hypothesis that TC-NER plays a pivotal role in the repair at polyG tracts in lung cancers, there was minimal transcriptional strand asymmetry for polyG motifs at genes that were not expressed or lowly expressed, and strong asymmetry for medium- and highly expressed genes (Mann–Whitney U p value < 0.001). This effect was also strongly-dependent on the length of the polyG motifs (Fig. 3d), (Kruskal–Wallis H-test with Bonferroni correction, p value < e − 0.5 for medium and high expression genes, p value > 0.05 for low expression genes).

Replication has also been reported to induce asymmetric mutation distributions between leading/lagging strands[3,4]. To exclude the possibility of replication strand orientation confounding our observations, we investigated whether indel transcriptional strand asymmetries at selected polyN motifs were related to leading and lagging replicative orientation. We found that replication strand orientation had limited effect on the observed indel transcriptional strand asymmetry of these tracts (Mann–Whitney U with Bonferroni correction, p value > 0.05 in all cases) (Supplementary Fig. 5a, b). This supports the role of

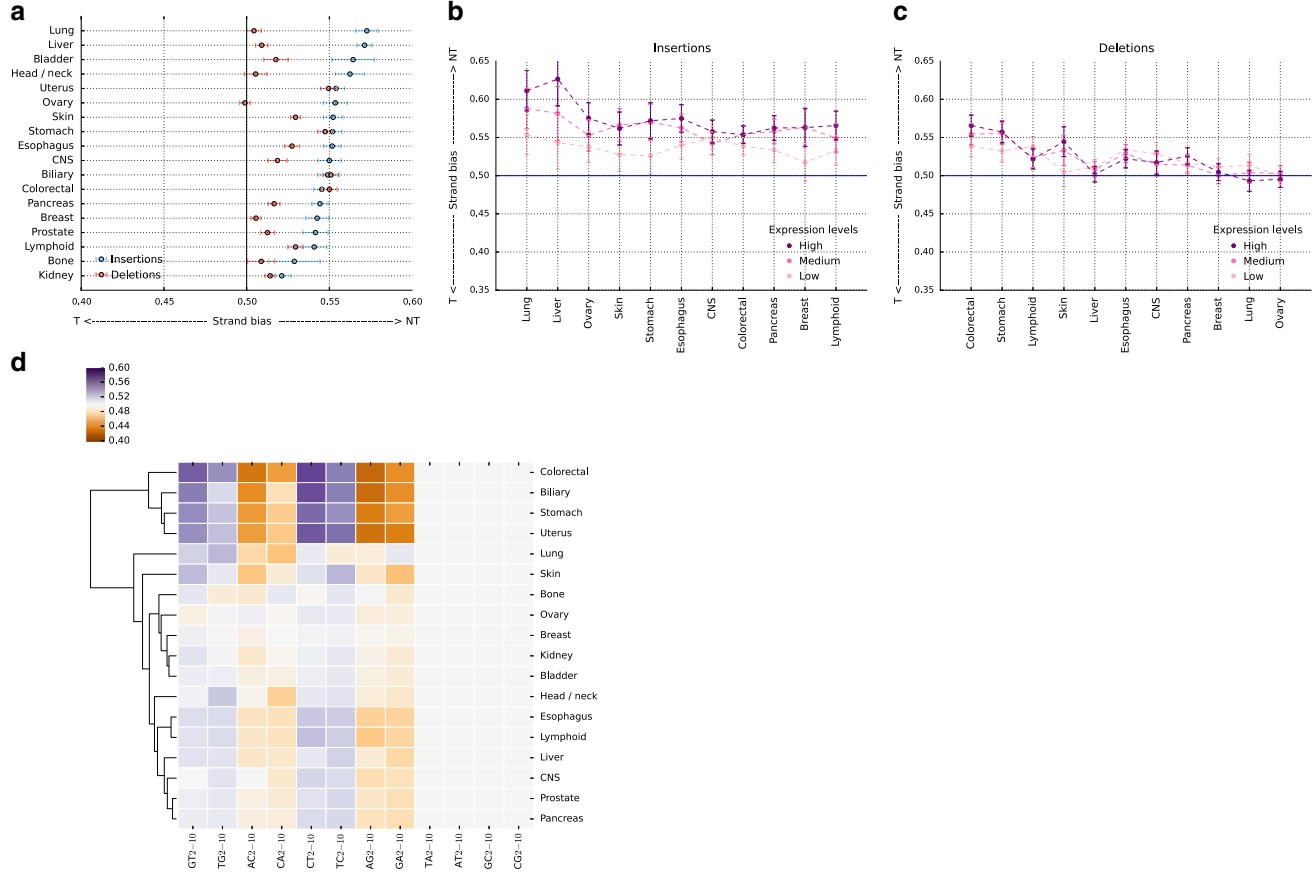

**Fig. 4 Transcriptional strand asymmetry at insertions and deletions. a** Transcriptional strand asymmetry of insertions and deletions at polyT tracts. Error bars represent standard deviation from bootstrapping with replacement. Both insertions and deletions displayed a strand asymmetry bias towards the non-template strand for polyT tracts across cancer types (binomial test with Bonferroni correction, $p$ value < 0.001 for insertions and $p$ value < 0.05 for deletions). **b** Transcriptional strand asymmetry occurring at polyT tracts according to level of gene expression for insertions. Mann–Whitney $U$ with Bonferroni correction, $p$ value < 0.001 when comparing low and high expression gene sets across all cancer types except skin, ovarian and lymphoid cancers ($p$ value < 0.05) and CNS ($p$ value > 0.05). **c** Transcriptional strand asymmetry occurring at polyT tracts according to level of gene expression for deletions. Mann–Whitney $U$ with Bonferroni correction, $p$ value < 0.001 when comparing low and high expression gene sets for skin and $p$ value < 0.05 for stomach and pancreatic cancers. **d** Hierarchical clustering displaying transcriptional strand asymmetries for indels overlapping dinucleotide motifs. Dinucleotide repeat tracts of up to five repeated units are displayed. Purple represents asymmetry towards the non-template strand, whereas orange represents asymmetry towards the template strand. In the dendrogram of cancers, biliary, uterus, colorectal and stomach cancers are more distant from the other cancers, and contain MSI samples, while lung cancers are also separable from other cancer types, further reinforcing our observations regarding the DNA damage and repair processes that contribute to the observed asymmetries. Across cancer types a non-template strand asymmetry preference was observed for TG, TC and CT motifs (binomial test with Bonferroni correction, $p$ value < 0.001) and for GT motifs (binomial test with Bonferroni correction, $p$ value < 0.05) and a template strand asymmetry for CA, GA and AG motifs (binomial test with Bonferroni correction, $p$ value < 0.001) and for AC motifs (binomial test with Bonferroni correction, $p$ value < 0.05).

transcription and excludes the influence of replication in the generation of indel transcriptional strand asymmetries.

**Insertions and deletions are differentially dependent on DNA repair pathways**. Next, we distinguished insertions from deletions at polyT and polyG tracts to find that transcriptional strand asymmetry differed between these classes of indels (Fig. 4a and Supplementary Fig. 6a, b). Insertions showed aggravated asymmetries at polyT tracts across all cancer types and were independent of MMR status, suggesting that mutagenesis associated with polyT tracts may be largely dependent on TC-NER (Wilcoxon signed-rank, $p$ value < e − 5) (Fig. 4a). By contrast, non-template strand bias of deletions at polyT tracts was restricted to tumour types that had a high incidence of MSI (biliary, colorectal, stomach and endometrial). Thus, mutagenesis that results in deletions is more heavily dependent on the MMR pathway.

To support this hypothesis, we investigated the relationship between transcriptional strand asymmetry for insertions and deletions at polyT motifs, and gene expression levels (Supplementary Table 2). Genes with higher expression levels displayed stronger transcriptional strand asymmetry of insertions at polyT tracts across all inspected cancer types (Fig. 4b, Mann–Whitney $U$ with Bonferroni correction, $p$ value < 0.05), implicating TC-NER, which is linked to expression levels[30]. However, a relationship between expression levels and transcriptional strand asymmetry of deletions at polyT tracts could only be observed for a subset of cancer types (Fig. 4c, Mann–Whitney $U$ with Bonferroni correction, $p$ value > 0.05) and the strand bias was less apparent. In contrast, at polyG motifs, we did not observe consistent associations between expression levels and transcription strand asymmetry of insertions or deletions across cancer types (Supplementary Fig. 6c, d, Mann–Whitney $U$ with Bonferroni correction $p$ value > 0.05), with the exception of lung cancers; this

we expected because of the influence of bulky adducts from tobacco carcinogens (Fig. 3d).

To provide further evidence for the role of TC-NER in the observed transcriptional strand asymmetry at polyT tracts for insertions relative to deletions, we reasoned that patients with defects in the TC-NER pathway would have indel patterns that should not demonstrate transcriptional strand bias because of defective NER. By contrast, patients with defects in global genome NER (GG-NER), may not manifest any changes in transcriptional strand bias. Tumour samples from these rare syndromes are however extremely difficult to obtain and systematic WGS data are not widely available to perform such analyses. We sequenced a cutaneous malignancy derived from a patient with an autosomal recessive DNA repair defect called xeroderma pigmentosum (XP). The patient was a compound heterozygote for the XPC gene, involved in GG-NER. Intriguingly, non-template strand bias was not observed for insertions at polyT tracts in this tumour, in contrast to what we had observed across cancer types (Fig. 4a, binomial test p value < 0.001). The numbers of insertions however were small in this single sample. We thus performed a down-sampling of the number of insertions for all cancer types to similar levels as the XP-mutant tumour to examine whether the difference in transcriptional strand asymmetry remained significant. The XP-deficient tumour consistently displayed decreased levels of non-template strand bias at insertions in comparison with all cancer types (Supplementary Fig. 7a, b), whereas for deletions there were no significant differences relative to other cancers (Supplementary Fig. 7c, d). A more robust assessment will be required in due course following collection of more XP-deficient tumours of the various XP proteins in the NER pathway. These tumours are extremely rare however and is beyond the scope of this paper for comprehensive collection and analysis.

**Transcriptional strand asymmetries at indels may be a general feature.** Finally, to understand whether our observations were restricted to mononucleotide tracts or if they could be a more generic mechanistic feature of indel mutagenesis, we attempted to explore other types of indels. The limitation is the difficulty in assigning other types of indels to specific strands. It was, however, possible to ascribe strandedness to dinucleotide repeat tracts. There were some caveats: palindromic GC/CG and AT/TA dinucleotides could not be oriented, and AA/TT/GG/CC dinucleotides were excluded because these are similar to mononucleotide polyA/T/G/Cs respectively. This left us with eight types of polydinucleotide repeat tracts that we could analyse (GT/TG/AC/CA/CT/TC/AG/GA), (Supplementary Fig. 8). Indeed, correcting for background asymmetries in the genome, we observed transcriptional strand asymmetries for several poly-dinucleotide repeat tracts (Fig. 4d). This was most marked amongst tumour types where MSI was prevalent (Fig. 4d and Supplementary Fig. 9, Mann–Whitney U tests with Bonferroni correction). Furthermore, strand asymmetry in insertions was stronger than in deletions (Wilcoxon signed-rank tests with Bonferroni correction, Supplementary Fig. 10a, b), with the exception of MSI tumours (Supplementary Fig. 10a, b). Thus, our findings appear to be applicable to motifs other than mono-nucleotide repeat tracts.

## Discussion
In this work, we have described a method to investigate transcriptional strand asymmetries for indels. Unexpectedly, we found biases in the distribution of mononucleotide repeat tracts in the reference genome at transcribed regions, and the bias is more pronounced for longer tracts. This bias needs to be

considered when exploring transcriptional strand asymmetries for indels overlapping mononucleotide repeat tracts.

Our analysis demonstrates strong and previously undescribed transcriptional strand asymmetries of indels. Our results implicate particular DNA repair pathways, namely TC-NER and MMR as contributing factors to the observed strand biases at indels (Figs. 3, 4). We further reveal that the formation of insertions is largely TC-NER-dependent, while the formation of deletions is additionally reliant on MMR, thus reinforcing how distinct mechanisms underpin the formation of different classes of indels.

## Methods
**Mutation calling.** Data were obtained from WGS cancers from ICGC under the project PanCancer Analysis of Whole Genomes[31]. They included 46 cancer projects from 21 organs. In total, 2575 WGS patients were analysed using the GRCh37 (hg19) reference assembly of the human genome.

Somatic indel calls were performed using three pipelines from four somatic variant callers. These were the Wellcome Sanger Institute pipeline, the DKFZ/EMBL pipeline and the Broad Institute pipeline[31], with somatic variant false discovery rate of 2.5%. Indel calling was performed by those algorithms and only indels called by at least two of the callers were analysed[31], therefore generating a conservative dataset (Supplementary Table 1). As a result, the false negative rate of indel detection could be higher than that of other methods, and of each pipeline separately, which implies that many indels present in the samples were not identified successfully. However, because of the large number of WGS tumour samples available, a sufficient number of indels remained (Supplementary Table 1). Finally, for a small subset of indels, the indel calls were visually examined using JBrowse Genome Browser[32], to inspect the number of reads reporting the indel, if the indel calls were biased towards the end of the sequencing reads or if there were other systematic biases between the normal and tumour sequencing reads; such biases could not be identified.

Bedtools intersect utility was used to measure overlap between indels and polyN tracts. The term overlap in this context refers to deleted bases occurring at any position across the entire length of the repeat or inserted bases occurring at any position across the length of the repeat and immediately before or after the repeat. Indel density was defined as the number of indel mutations for a given number of bases.

The distance between each pair of consecutive indels was calculated per patient. Indels in different chromosomes were excluded since we could not define their pairwise distance. The same analysis was also performed separately for insertions and deletions to generate Supplementary Fig. 1a.

Indels from HAP1 cells with MutS homolog 6 (MSH6) knock-out were obtained from Ref. [29]. Indels from cells exposed to various PAHs, namely benzo[a]pyrene [0.39 and 2 μM] and benzo[a]pyrene diolepoxide [0.125 uM], were obtained from Ref. [33] to examine transcriptional strand asymmetry at indels overlapping polyN motifs in experimental settings.

Substitution calling was performed using four somatic mutation-calling algorithms, with mutation calls being shared by at least two algorithms[31]. For lung cancers, C > A substitutions were examined with respect to transcriptional strand asymmetries at polyG tracts and replication timing (Supplementary Fig. 6e).

Mutational enrichment at MSI over MSS samples was defined as:

Ratio: (proportion of indels overlapping polyA/T at MSI samples)/(proportion of indels overlapping polyA/T at MSS samples).

**Transcriptional strand asymmetries at the human genome.** Gene annotation from Ensembl was followed[34] and genes were downloaded from Biomart (http://grch37.ensembl.org/biomart/martview/c1d06f3affb6260c0cd7147bb4c3b6a8) using Gene start and Gene end to define genes and filtering by only including protein-coding genes and we also selected the attributes Strand and Gene Name. BEDTools utilities v2.21.0 were used to manipulate genomic files and intervals[35]. GC-skew is a measure of bias in the number of Gs or Cs between the template and non-template strands. GC-skew was calculated as $(G - C)/(G + C)$ for windows of 100 bp around the TSS and TES. Similarly, AT-skew was calculated as $(A - T)/(A + T)$ for windows of 100 bp around the TSS and TES (Supplementary Fig. 2a, b).

Genes in the positive and negative orientations were separated to determine the direction of gene transcription. Scripts were written in python to identify non-overlapping polyN motifs of size 1–10 bp as well as dinucleotide motifs of length 2–10 bp genome-wide and orient them in terms of transcription direction at genic regions.

Template motifs were the motifs in: (i) positive gene orientation and negative genome strand, (ii) negative gene orientation and positive (reference) genome strand. Non-template motifs were the motifs in: (i) positive gene orientation and positive genome strand (reference), (ii) negative gene orientation and negative genome strand. Bedtools intersect utility was used to calculate motif occurrences in template and non-template strands across genic regions.

To investigate the effect of the distance from the TSS and the TES across the gene length, for genes with unequal gene length, we divided each gene into ten genomic bins of equal size. Also, two additional bins upstream from the TSS and

two bins downstream of the TES, each 10 kB in size, were added. Then, we calculated the frequency and the strand asymmetry bias of polyN motifs in each genic bin (Fig. 2a and Supplementary Fig. 3). In particular, we calculated the density of polyNs at a bin as the number of polyNs over the total number of bases at that bin. However, we derived the enrichment of polyNs at the bin by comparing the ratio of the density at the bin against the density across all bins.

Relative enrichment of a polyN tract at a bin was calculated as:

Enrichment = (density of polyN motif at bin)/(density of polyN motif across all bins).

Strand asymmetry bias was calculated as:

(motif occurrences at non-template strand)/(motif occurrences at template strand).

The distribution of polyN motifs at the template and non-template strands relative to the TSS and the TES was calculated with bedtools *intersect* command using the gene orientation approach described earlier to generate (Fig. 2c, d) (Supplementary Fig. 3a–g). Bootstrapping using random sampling of genes with replacement was performed from which the standard deviation of the strand asymmetry bias was calculated. For Fig. 2c, d the interval used was 100 bp and error bars represent standard error from bootstrapping with replacement (1000-fold).

**Template/non-template strand asymmetries in cancer**. The numbers of indels overlapping motifs found in the template or non-template strands were obtained using the bedtools *intersect* command. Strand bias was calculated for the vector of genes, reporting the number of polyN motif occurrences and the number of overlapping motifs as:

A = (indels overlapping motif at non-template)/(motif occurrences at non-template)

B = (indels overlapping motif at template)/(motif occurrences at template)

Strand bias = A/(A + B)

with motifs representing polyN repeat tracts of size 2–10 bp and dinucleotide repeat tracts of 1–5 repeated units, at genic regions (Figs. 3a–d, 4a–d).

We performed bootstrapping with replacement, randomly selecting the indels overlapping motifs at template and non-template strands from each randomly selected gene, for equal number of genes in multiple iterations, from which we calculated the standard deviation for the strand bias.

MMR-deficient samples were identified using genome plots and mutational signature profiles of each patient for stomach, biliary, uterus and colorectal tumours. Subsequently, transcriptional strand asymmetry levels at indels overlapping polyT tracts were compared between MSS and MSI samples to investigate the role of MMR in transcriptional strand asymmetries (Fig. 3b and Supplementary Fig. 9).

**RNA-seq analysis**. For the comparative analysis between expression levels and transcriptional strand asymmetry, cell-of-origin cell lines, where available, were used from Roadmap Epigenomics project[36] (Supplementary Table 2). For each cell line, genes were grouped in expression level quantiles, namely 'low', 'medium' and 'high' based on the associated RPKM gene expression values. The groups were defined using the 33rd and 66th percentiles from the RPKM gene expression values for protein-coding genes.

Transcriptional strand asymmetry at indels overlapping polyN motifs was investigated in relation to gene expression levels to generate Figs. 3d, 4b, c and Supplementary Fig. 6c, d.

For lung cancer, using cell-of-origin RNA-seq data (IMR-90) from Roadmap Epigenomics project[36], polyG tracts were grouped according to their length to investigate if the length of polyG tracts was associated with transcriptional strand asymmetry at indels across the gene expression quantiles (Fig. 3d).

**XPC dataset**. A cutaneous malignancy derived from a patient with an autosomal recessive DNA repair defect called XP mutation was obtained from Ref. [37]. The patient was a compound heterozygote for the XPC gene. We performed the non-template strand asymmetry analysis for insertions and deletions overlapping polyT tracts (Supplementary Fig. 7a–d). To control for the lower number of indels in the patient we randomly selected equal number of insertions and deletions in each cancer type, weighting for the observed transcriptional strand asymmetry at polyT tracts in each cancer type and we compared the transcriptional strand asymmetry profile of the XP sample to that of each cancer type and calculated the associated z score and p value from 10,000-fold bootstrapping this process for each cancer type (Supplementary Fig. 7a–d).

**Replication timing analysis**. Repli-Seq data for IMR-90 cell line were obtained from The ENCODE Project Consortium[38] and replication domains were generated using the observed Repli-seq signal[4]. Genes were grouped across five replication timing quantiles and transcriptional strand asymmetry at indels overlapping polyG tracts within transcribed regions was calculated for each quantile (Supplementary Fig. 6e). The same type of analysis was performed for lung cancer C > A (G > T) substitutions to investigate the contribution of replication timing to the levels of transcriptional strand asymmetries at substitutions and indels overlapping polyG motifs of 2–10 bp length (Supplementary Fig. 6e).

Leading and lagging orientation of the replication machinery across the human genome was inferred for MCF-7 cell line with Repli-seq data by using the finite difference approximations of second and first derivatives[4]. Subsequently, polyN motifs were separated into those occurring in the leading orientation and those in the lagging orientation. The indel transcriptional strand asymmetry analysis was performed separately for polyT and polyG motifs occurring at the leading and lagging orientations therefore controlling for the effect of replication orientation (Supplementary Fig. 5).

Statistical analyses across the manuscript were performed in python with packages 'math', 'scipy', 'pandas', 'scikit-learn' and 'numpy'. Figures across the manuscript were generated in python using packages 'matplotlib', 'seaborn' and 'pandas'.

**Reporting summary**. Further information on research design is available in the Nature Research Reporting Summary linked to this article.

## Data availability
Relevant files including mutation data count tables can be found here: https://data.mendeley.com/datasets/kdywxnn729/3

Primary mutation data were obtained from ICGC under the project PanCancer Analysis of Whole Genomes (PCAWG)[31]. A cutaneous malignancy derived from a patient with an autosomal recessive DNA repair defect called Xeroderma Pigmentosum (XP) mutation was obtained from Ref. [37]. Indel mutational profiles of non-cancerous human cells exposed to various polycyclic aromatic hydrocarbons (PAHs) including benzo[a]pyrene [0.39 and 2 μM] and benzo[a]pyrene diolepoxide [0.125 μM] were derived from ftp://ftp.sanger.ac.uk/pub/cancer/Zou_et_al_2017 and experimentally-generated mutation patterns from CRISPR-Cas9 knockouts of a human cancer cell line for *MSH6* were derived from https://data.mendeley.com/datasets/m7r4msjb4c/2. All data is available from the authors upon reasonable request.

## Code availability
All associated code has been deposited in https://data.mendeley.com/datasets/kdywxnn729/3 and is available from the authors upon reasonable request.

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

## Acknowledgements
M.H. is supported by the Wellcome Trust Sanger Institute core grant. S.N.Z. is funded by a CRUK Advanced Clinician Scientist Award (C60100/A23916) and a CRUK Grand Challenge Award (C60100/A25274). J.J. is funded by the Swiss National Science Foundation (31003B-170267).

## Author contributions
I.G.-S., M.H. and S.N.-Z. conceived the concepts and analytical framework and drove the intellectual exercise. I.G.-S. wrote the code for analysing and presenting the data. S.E.M. generated the XPC-deficient tumour data under the supervision of S.N.-Z., I.G.-S., M.H. and S.N.-Z. wrote the manuscript with the help of G.K., S.E.M. and J.J.

## Competing interests
S.N.Z. has patent applications with the UK IPO. S.N.Z. is also a consultant for Artios Pharma Ltd, Astra Zeneca and the Scottish Genomes Partnership. The other authors declare no competing interests.
