## [Peer Review File · Nature Communications]

Reviewers' comments:

Reviewer #1 (Remarks to the Author):

This paper analyzes indels, focusing mostly on indels on polyN track, and reports that there is a transcription strand bias in the somatic indels in tumors and associates this fact to the role of NER-TC in their repair. Although the analysis is of interest, more conclusive evidences are needed to support the main conclusions of the paper.

Major issues:

- Replication fork progression is significantly co-oriented with the transcription, which means that replication strand and transcription strand overlap in many genes. Then, there is the possibility that the transcription strand bias in polyN indels reported in the paper actually reflects a replication strand bias. If so, the conclusion that it is NER-TC the responsible of this bias could be incorrect. Can the authors rule out that this is the case? They should redo analyses accounting for the confounding of replication strand bias to see if the transcriptional strand bias is still maintained.

- How the authors think that their results relate to those of this paper? Tubbs et al., Dual Roles of Poly(dA:dT) Tracts in Replication Initiation and Fork Collapse. Cell 2018

- To show if the transcriptional strand asymmetry in polyN indels is truly related to TC-NER, authors could analyze indels from samples or cells deficient in NER to see if the asymmetry is still there or not.

- Why polyG indel transcription asymmetry is not associated to expression (Fig S4C, D)? This goes against the hypothesis that TC-NER is involved in repairing polyG indels, right? How do authors explain this?

- Figure 2A. The enrichments of polyN motif in different bins may simply or partly reflect the sequence composition of the gene. Probably the first bin is enriched in GCs and that's why it appears enriched in polyGs and polyCs compared to the rest of regions of the gene. The authors should check if the polyN enrichment is due to enrichment of Ns or really enrichment of polyNs?

- Figure 2C and D. Similarly there is a strand bias in sequence content which could, at least partly, explain the bias in polyGs and polyTs by transcription strand. The authors should clarify this.

Minor:

Do the authors have any hypothesis or potential explanation on why the ratio of deletions to insertions change by cancer type? For instance, why it is higher in ovary?

(Zou et al. in press) (page 17) I guess it refers to (Zou et al. 2018)

The quality of the figures can be improved. For example, in Fig 1b, there is too much space. It is also missing the number of samples of each tumor type, and a unified color for each of the tumor types could also be helpful.

It would also be interesting to perform an NMF extraction using channels similar to those used in the PCAWG Mutational Signatures main paper but with transcribed and non transcribed channels (similar to the usual 192 channels for strand asymmetry in SNVs). This might help on deciding whether the transcription bias (if any) is specific to a mutational process or a general feature.

Reviewer #2 (Remarks to the Author):

In this manuscript, the authors attempt to correlate insertion and deletion frequency, as detected in cancer genome sequencing, with sequence motifs, in particular short mono- and dinucleotide repeats, within transcribed regions. Their analysis suggests that there are transcription strand and sequence-context specific mechanisms that alter the likelihood of insertions/deletions. Specifically, nucleotide excision repair and mismatch repair are suggested to influence this strand- and sequence-bias.

The question posed in this paper is interesting, ie: "does transcription affect the occurrence of insertion/deletion mutations?". Using repeats as a way to "orient" the sequence is also a reasonable approach. The presented results look interesting. However, I am not yet 100% convinced of the interpretation of the data, because several parts of the analysis are not very well described and some inferences seem a bit premature. Below, I am listing several points which I feel are pressing. There are other parts of the analysis which I'm not sure about but cannot really decide on until I have a better understanding of the entire methodology.

___Major issues: ___

1. Neither the data nor the computer code used to analyse the data are accessible. This makes it impossible to replicate the results, or to validate whether there are issues in the analysis. I feel this is a non-starter for a computational biology paper. Specifically, I would like to see

- URL(s) or accession number(s) for the mutation data
- Uploads of code and other necessary files for the pre-processing and the statistical analysis on FigShare, Zenodo or similar
- A bed file or similar with the exact indel coordinates and definitions used

2. It is unclear to me whether the definition of "gene" used here refers to protein coding genes, or also to pseudogenes, lncRNAs, tRNA etc. The methods say: "Gene annotation from Ensembl was followed (Aken et al. 2016)" - Could the authors clarify, reference the URL, and include the gene definition with the code?

3. It is unclear how the authors treated exons and introns. Were both exons and introns considered jointly as the "gene"? Is the insertion/deletion strand asymmetry restricted to exons/introns only?

4. I fail to grasp the bootstrapping approach described under "Template / Non-template strand asymmetries in cancer" in the methods:

> "We performed bootstrapping with replacement, randomly selecting the indels overlapping motifs at template and non-template strands from each randomly-selected gene, for equal number of genes in multiple iterations, from which we calculated the standard deviation of enrichment." I would instead suggest to generate an equal number of indels with random locations, that seems easier and cleaner.

___Other points: ___

1. How is an "overlap" between a repeat and an insertion/deletion defined? Do the authors look for overlaps between the breakpoint(s) of the insertion/deletion and the repeat, or do they consider the whole length of the inserted/deleted region?

2. The authors use the expression "density" a lot. Could they clarify how "indel density", or "polyN density in a bin" are defined? I'm assuming they mean density as a synonym for "frequency", i.e. "number of observations divided by all possible observations"? But for e.g. Supp Fig 1C, how is "density of indels at polyA/T tracts" defined? Do they take the average number of indels per poly-

A/T tract, divided by the length of the tract? Or relative to the number of poly-A/T tracts considered? Or both?

3. Is the relationship between indel frequency and strand dependent on the repeat size?

Reviewer #3 (Remarks to the Author):

The manuscript by Serena Nik-Zainal and colleagues is devoted to an interesting and relatively unexplored subject of analyzing indels taking into account their relative orientation to the nearest transcribed gene. Authors use this method to discover transcriptional strand asymmetry in the distribution of mononucleotide repeat tracks in the human genome. They also observe same transcriptional asymmetry across 21 cancer types encompassing a curated collection of 2,575 whole-genome sequencing data sets. To explain the source of underlying asymmetry, the authors used several carefully chosen conditions, including cancer samples with deficient mismatch repair, and non-cancerous human cells exposed to conditions mimicking carcinogenic effects of tobacco smoke. Nik-Zainal and colleagues also use gene expression from cell-of-origin for different cancers to study impact of expression levels (known to be correlated with transcription-coupled nucleotide excision repair) to support their hypothesis that transcription-coupled nucleotide excision repair plays an important role in the repair of polyG motifs in lung cancer, for which they found strong transcriptional bias for creating overlapping indels. The manuscript is written in a clear and an engaging manner.

What is disappointing, though, is that even though the work presented is entirely computational, with few exceptions the analysis is not performed according to good computational standards. Namely, instead of using statistical tests and p-values to assess enrichments or significance of the results, a reader is asked to “visually appreciate” the presented data. The data is also often presented as ratios, which can accumulate much higher error rates than individual numbers used for their calculations. These shortcomings are not allowing a reader to assess which of the presented results are truly significant and whether all conclusions drawn by the authors are supported by the data they present.

Specific remarks:

Figure 1, all panels (a,b,c) would benefit from discussion of statistical significance of difference in discussed distributions. Comparing all possible pairs is not feasible due to the high number of different cancer types considered, but authors could e.g. compare distribution for each cancer with the total distribution for all cancer types considered or do pairwise comparisons for cancer types adjacent to each other in the figures.

Figure 2A. Current visualization does not give any information if the observed differences are significant. If distribution of ratios is close to normal, Z-transform will be helpful, i.e. expressing ratios in terms of their distance from the mean, expressed in standard deviations. For example, it will be more helpful to know that deep blue denotes 5 standard deviations above the mean than 1.4 mean, since the latter result may or may not be significant, while the former would be, even after applying correction for multiple hypothesis testing, such as the Bonferroni correction (here it will amount to multiplying p-values calculated for individual bins by the total number of bins). The reason multiple hypothesis correction (e.g. Bonferroni) should be applied is the authors consider 448 bins, so we expect to see 22 events (bins) with individually calculated p-value ≤ 0.05 (i.e. 1 in 20), which corresponds to the distance of 1.96 standard deviations from the mean. On the other hand, we will randomly observe values 5 sigmas away from the mean only once in 1,744,278 times, so such values will be clearly significant, even after correction for multiple hypothesis testing.

Figure 2C, D. Error bars need to be added, it is also not stated in which intervals density is computed

Figure 3D. Error bars need to be added.

Line 170. The test used to computing p-value needs to be stated.

Lines 188 -192. P-value of the result should be computed.

Lines 194-200. No information is given on what criteria were used for dividing genes into highly- and lowly-expressed. How many genes were in each group? Fig. 3D show "polyG motifs at genes" – does it also includes previously used 4 bins outside of gene body?

Line 232 "specific stRands", not "stands"

Fig 4 B, C Error bars are necessary to assess results

Fig. 4 B and C shows the dependence of transcriptional strand asymmetry in indels for all cancer types. I understand before expression levels were analyzed for each cancer cell type separately, using cell-of-origin. How the genes were divided here into these three groups globally based and their varying expression levels in different conditions? Why two groups used before (high and low) were not used?

The same consideration (need for error bars and statistical assessment of enrichments) apply to some supplementary figures.

If these problems are satisfactorily addressed, I think the manuscript will be suitable for publication in Nature Communications.

Response to Reviewers' comments

Reviewer #1 (Remarks to the Author):

This paper analyzes indels, focusing mostly on indels on polyN track, and reports that there is a transcription strand bias in the somatic indels in tumors and associates this fact to the role of NER-TC in their repair. Although the analysis is of interest, more conclusive evidences are needed to support the main conclusions of the paper.

We thank the reviewer for the positive response and queries which are addressed below. These have substantially improved the manuscript. Thank you.

Point 1.1

Major issues:

- Replication fork progression is significantly co-oriented with the transcription, which means that replication strand and transcription strand overlap in many genes. Then, there is the possibility that the transcription strand bias in polyN indels reported in the paper actually reflects a replication strand bias. If so, the conclusion that it is NER-TC the responsible of this bias could be incorrect. Can the authors rule out that this is the case? They should redo analyses accounting for the confounding of replication strand bias to see if the transcriptional strand bias is still maintained.

Thank you for the suggestion. A very important confounder to exclude. We performed additional analyses to show that the observations are not confounded by replication.

PolyT and polyG tracts were orientated relative to direction of replication fork progression (leading or lagging). Transcriptional strand asymmetry was sought in each orientation. The direction of replication did not significant affect levels of transcriptional strand asymmetry (Mann-Whitney U tests). These results are now included as Supplementary Figure 5.

A

B

Supplementary Figure 5: Transcriptional-strand asymmetry of indels at polyT and polyG tracts controlling for the effect of replication direction. Transcriptional strand asymmetry at A. polyT tracts and B. polyG tracts separated by leading and lagging replicative strands, to consider potential confounding due to replication direction. Repli-seq data were derived from MCF-7 cell line. For polyG and polyT tracts the replication direction did not significantly affect the transcriptional strand asymmetry levels (Mann-Whitney U test Bonferroni corrected, p-value>0.05 for all cancer types). X-axis shows transcriptional strand bias (non-template) / (template + non-template), y-axis shows different tissue types. Repli-seq data were derived from MCF-7 cell line. Error bars represent standard deviation from bootstrapping with replacement.

Point 1.2

- How the authors think that their results relate to those of this paper? Tubbs et al., Dual Roles of Poly(dA:dT) Tracts in Replication Initiation and Fork Collapse. Cell 2018

Tubbs et al., showed that replication initiation zones are flanked by asymmetrical poly(dA:dT) tracts and are usually found between transcribed genes. Long poly(dA:dT) sites (>20bp) are often at fragile sites and have frequent double-strand breaks. In addition, there is an asymmetrical pattern between the poly(dA:dT) tract orientation and the break ends of the double-strand breaks.

Our findings are not inconsistent with the results of Tubbs et al. Indeed, we observe high mutation rates at poly(dA:dT) sites and the mutability is correlated with the length of the tract. However, we do not focus at long poly(dA:dT) tracts (>20bp) as in Tubbs et al. but rather at small poly(dA:dT) tracts (<11bp) and instead of double-strand breaks we concentrate on small indels between 1-100bp length in transcribed genic regions. We also detect asymmetries of poly(dA:dT) tracts but we find that transcription orientation rather than replication orientation plays a central role. However, it should be noted that we are looking at different types of polyN tracts.

Interestingly, in the discussion Tubbs et al. suggest that as in vitro studies have shown DNA polymerase strongly pauses when the template contains poly(dT) but not poly(dA) tracts. A similar mechanism could be in place for the transcribed regions, in which RNA polymerase pausing or speed could be associated with template and non-template orientation of polyT tracts, which would explain the strong strand asymmetry that we observe.

Point 1.3

- To show if the transcriptional strand asymmetry in polyN indels is truly related to TC-

NER, authors could analyze indels from samples or cells deficient in NER to see if the asymmetry is still there or not.

Thank you for the suggestion. We obtained a cancer sample from a patient who has xeroderma pigmentosum, that is an inherited autosomal recessive disorder of the XPC gene. These conditions are very rare and to obtain such a sample (with adequate quality for whole genome sequencing - because to see insertion/deletion patterns, one requires genome scale data) is unusual. Most samples have been formalin-fixed in the past. While this was a helpful exercise, it was also quite limited which we explain below.

In-keeping with our hypothesis of the role played by a component of nucleotide-excision repair in contributing towards the observed non-template strand bias in human cancers, we find that in the XP-mutant tumor, insertions at polyT tracts fail to display non-template strand asymmetry (Binomial test: $p\text{-value} < 0.001$)

We also do not observe significant strand asymmetry for deletions, as expected and seen in human cancers.

However, there are a relatively low number of insertions in this sample and given the rarity of the disease, it is unlikely that we will be able to source a large number of samples for WGS to be adequately powered. To deal with the problem of having a low number of insertions in the single tumor, we randomly selected an equivalently low number of insertions at polyT tracts for each cancer type, weighting for their frequency at template and non-template polyT tracts and performed multiple simulations ($n=10,000$).

Next, we calculated the p-value and z-score by comparing the strand asymmetry of the sample of the XP patient to those derived from bootstrapping per cancer type (Supplementary Figure 7a-d). We found that even after this down-sampling exercise, all cancer types displayed increased non-template strand asymmetry in comparison to the XP patient for insertions.

These findings suggest that XP proteins contribute towards non-template strand bias observed in insertions although the numbers are low and will require more samples to be collected across different XP proteins to fully understand the picture.

Point 1.4

- Why polyG indel transcription asymmetry is not associated to expression (Fig S4C, D)? This goes against the hypothesis that TC-NER is involved in repairing polyG indels, right? How do authors explain this?

Apologies for the misunderstanding.

For lung cancers, polyG indel transcriptional strand asymmetry is associated with expression and is even shown to correlate with length of the polyG tract.

Please see Figure 3D. These calculations are performed for lung cancers because they have the highest levels of indel transcription asymmetry of polyGs, in-keeping with the notion that G-adducts formed by the PAHs in tobacco-smoke are repaired by TCR.

Figures S6C and S6D show the relationships across all cancer types. However, polyG-indel asymmetry is weak and variable in most cancer-types apart from lung, which shows strong non-template strand asymmetry (Figure S6B). That is why it looks like there is no relationship with expression (because there is no asymmetry) for many cancer types. However, for lung cancers where we know that the damage is due to adducted guanine which is dependent on TCR for repair, the association with expression is strong (Figure 3D).

Point 1.5

- Figure 2A. The enrichments of polyN motif in different bins may simply or partly reflect the sequence composition of the gene. Probably the first bin is enriched in GCs and that's why it appears enriched in polyGs and polyCs compared to the rest of regions of the gene. The authors should check if the polyN enrichment is due to enrichment of Ns or really enrichment of polyNs?

Yes, we know that there are variable enrichments of GC/AT through a gene. But Figure 2A shows relative enrichments of all types of polyN tracts of all lengths. It is not of simply any N, it has to be in the motif that is described in the legend.

In other words, a "GGGG" motif is a polyG motif of G=4.

We do not then double-count. We only count a single occurrence of each polyN tract, as its longest length. This means that a "GGG" motif will not be counted as "GG" twice. As a result, the enrichment of polyGs of a specific length in the first bin is not driven by the enrichment of random Gs, or double-counting in the same bin. Each type of polyN tract is an independent, non-double-counted motif.

Point 1.6

- Figure 2C and D. Similarly there is a strand bias in sequence content which could, at least partly, explain the bias in polyGs and polyTs by transcription strand. The authors should clarify this.

The distribution of mononucleotide Gs and mononucleotide Ts is shown in Figures 2C-D, indicating that strand asymmetry is exacerbated at longer polynucleotide tracts and is therefore not explained simply by the sequence content. As in point 1.5, we do not double-count polyN tracts.

Point 1.7

Minor:

Do the authors have any hypothesis or potential explanation on why the ratio of deletions to insertions change by cancer type? For instance, why it is higher in ovary?

The higher variance in length observed in deletions in comparison to insertions (Levene's test, p -value <0.05) can be attributed in the contribution of microhomology-mediated end-joining to the formation of deletions. Ovarian cancers tend to have a lot of microhomology-mediated deletions which tend to be larger in motif size. The mechanisms underpinning deletions are different to insertions as well, so there is no reason for the variation in size of insertion motif to be mirrored by deletions.

(Zou et al. in press) (page 17) I guess it refers to (Zou et al. 2018)

Thank you. We have modified this reference to (Zou et al. 2018).

Point 1.8

The quality of the figures can be improved. For example, in Fig 1b, there is too much space. It is also missing the number of samples of each tumor type, and a unified color for each of the tumor types could also be helpful.

Thank you for the suggestions. We have made modifications to Figure 1 that we hope have improved the general presentation of the manuscript. For figure 1b we have assembled insertions and deletions together in a single panel, colored with two different colors unified across cancers. Also, additional changes include adding the number of samples in figure 1a, the addition of a panel regarding the z-scores obtained when comparing the ratio of deletions to insertions across patients to individual cancer types (Supplementary Figure 1c). Finally, we have included additional statistics found in the figure legend comparing the difference in variance between insertions and deletions and differences between cancer types as well as providing a statistical evaluation of the difference in frequency of insertions and deletions.

Point 1.9

It would also be interesting to perform an NMF extraction using channels similar to those used in the PCAWG Mutational Signatures main paper but with transcribed and non transcribed channels (similar to the usual 192 channels for strand asymmetry in SNVs). This might help on deciding whether the transcription bias (if any) is specific to a mutational process or a general feature.

With the greatest respect, we are inclined to decline this suggestion.

The PCAWG indel channels are neither validated nor widely used. Incorporating it into our paper would be to perpetuate a principle that is currently unsubstantiated nor accepted by peer review. The indel signatures have 83 channels, many of which never report any information and are not contributing to biological meaning.

This manuscript is not an exposition of the PCAWG mutational signatures paper. We do not think that it is wise to be caught up in the vagaries of a manuscript when it has not been through review yet (after 1 year). Thus, with respect, we do not feel that this is

such a good idea. We do apologise for disagreeing on this score – it seems an unnecessary complication.

In all however, we would like to thank Reviewer 1 for the very helpful suggestions that we feel have helped our manuscript.

Reviewer #2 (Remarks to the Author):

In this manuscript, the authors attempt to correlate insertion and deletion frequency, as detected in cancer genome sequencing, with sequence motifs, in particular short mono- and dinucleotide repeats, within transcribed regions. Their analysis suggests that there are transcription strand and sequence-context specific mechanisms that alter the likelihood of insertions/deletions. Specifically, nucleotide excision repair and mismatch repair are suggested to influence this strand- and sequence-bias.

The question posed in this paper is interesting, ie: "does transcription affect the occurrence of insertion/deletion mutations?". Using repeats as a way to "orient" the sequence is also a reasonable approach. The presented results look interesting. However, I am not yet 100% convinced of the interpretation of the data, because several parts of the analysis are not very well described and some inferences seem a bit premature. Below, I am listing several points which I feel are pressing. There are other parts of the analysis which I'm not sure about but cannot really decide on until I have a better understanding of the entire methodology.

Point 2.1

___Major issues:___

1. Neither the data nor the computer code used to analyse the data are accessible. This makes it impossible to replicate the results, or to validate whether there are issues in the analysis. I feel this is a non-starter for a computational biology paper. Specifically, I would like to see
 - URL(s) or accession number(s) for the mutation data
 - Uploads of code and other necessary files for the pre-processing and the statistical analysis on FigShare, Zenodo or similar
 - A bed file or similar with the exact indel coordinates and definitions used

We apologise for this thus far.

We have placed all mutation counts and code here:

<https://data.mendeley.com/datasets/kdywxnn729/3> and in

https://github.com/IliasGeoSo/Transcriptional_strand_asymmetry_Indels.

Point 2.2

2. It is unclear to me whether the definition of "gene" used here refers to protein coding genes, or also to pseudogenes, lncRNAs, tRNA etc. The methods say: "Gene annotation from Ensembl was followed (Aken et al. 2016)" - Could the authors clarify, reference the

URL, and include the gene definition with the code?

We apologise for this lack of clarity.

We referred only to protein-coding genes as defined by the Ensembl annotation. We used Biomart selecting "Gene_type = protein coding" and used "gene start" and "gene end" to define genes. We have updated the methods section to provide this clarification.

Point 2.3

3. It is unclear how the authors treated exons and introns. Were both exons and introns considered jointly as the "gene"? Is the insertion/deletion strand asymmetry restricted to exons/introns only?

The analysis was performed across the gene body including both exons and introns as the transcribed footprint encompasses both exons and introns (~40% of the genome). It would not be possible to assess transcriptional strand asymmetry in intergenic sequences.

Whether there is replication strand asymmetry or not is a separate question and one raised as a potential confounder by Reviewer 1, which we believe that we have addressed in point 1.1.

Point 2.4

4. I fail to grasp the bootstrapping approach described under "Template / Non-template strand asymmetries in cancer" in the methods:

> "We performed bootstrapping with replacement, randomly selecting the indels overlapping motifs at template and non-template strands from each randomly-selected gene, for equal number of genes in multiple iterations, from which we calculated the standard deviation of enrichment."

I would instead suggest to generate an equal number of indels with random locations, that seems easier and cleaner.

Bootstrapping was performed by randomly selecting equal number of genes and replicating our analysis across them, allowing us to calculate the standard deviation of the asymmetry score.

However, we took into consideration an alternative approach as described by the reviewer to provide evidence that our results are consistent. We have generated an equal number of indels per cancer type to replicate the observed asymmetries observed from the bootstrapping method described in the manuscript. For each indel a random indel of same length, in the same chromosome, randomly located within 2.5kB from the original indel was generated, not overlapping with blacklisted regions for which we do not perform mutational calling and of course also excluding regions falling outside the chromosome start and end.

Our results are largely unchanged following this additional method and are provided below for polyTs and polyGs.

The relevant scripts have been uploaded in Mendeley.

Point 2.5

___Other points:___

1. How is an "overlap" between a repeat and an insertion/deletion defined? Do the authors look for overlaps between the breakpoint(s) of the insertion/deletion and the repeat, or do they consider the whole length of the inserted/deleted region?

We now provide a clearer description of this in the materials and methods section. "Overlap" is defined as any deleted bases occurring at any position across the entire length of the repeat OR inserted bases occurring at any position across the length of the repeat and immediately before or after the repeat.

Point 2.6

2. The authors use the expression "density" a lot. Could they clarify how "indel density", or "polyN density in a bin" are defined? I'm assuming they mean density as a synonym for "frequency", i.e. "number of observations divided by all possible observations"? But for e.g. Supp Fig 1C, how is "density of indels at polyA/T tracts" defined? Do they take the average number of indels per poly-A/T tract, divided by the length of the tract? Or relative to the number of poly-A/T tracts considered? Or both?

Indel density is defined as the number of indel mutations over a given number of bases. In general, it is a preferred description and a more precise representation than frequency, because one is reporting the number of indels per Mb or per kb.

The confusion likely stems from figure 2A where we calculate the density of polyNs for each bin, as the number of polyNs over the total number of bases in that bin. Regardless, the principle still stands, that a density reports the total number of any feature given a number of bases (at any scale).

Point 2.7

3. Is the relationship between indel frequency and strand dependent on the repeat size?

Indeed, strand asymmetry is aggravated at longer polyT tracts across all cancers (Kruskal-Wallis H test, Bonferonni corrected, p-value<0.001 across cancer types) and for polyG tracts in lung and liver cancers (Kruskal-Wallis H test, Bonferonni corrected, p-value<0.001) and for pancreatic cancers (Kruskal-Wallis H test, Bonferonni corrected, p-value<0.05), (Supplementary Figure 4). For lung cancers we captured the relationship between the length of polyG tracts and the strand asymmetry as depicted in Figure 3D and further backed with the relevant statistical tests. However, due to lower number of mutations overlapping polyG tracts inferring these relationships remained harder for many cases and we think larger sample sizes in the future will help further extend these claims.

Reviewer #3 (Remarks to the Author):

The manuscript by Serena Nik-Zainal and colleagues is devoted to an interesting and relatively unexplored subject of analyzing indels taking into account their relative orientation to the nearest transcribed gene. Authors use this method to discover transcriptional strand asymmetry in the distribution of mononucleotide repeat tracks in the human genome. They also observe same transcriptional asymmetry across 21 cancer types encompassing a curated collection of 2,575 whole-genome sequencing data sets. To explain the source of underlying asymmetry, the authors used several carefully chosen conditions, including cancer samples with deficient mismatch repair, and non-cancerous human cells exposed to conditions mimicking carcinogenic effects of tobacco smoke. Nik-Zainal and colleagues also use gene expression from cell-of-origin for different cancers to study impact of expression levels (known to be correlated with transcription-coupled nucleotide excision repair) to support their hypothesis that transcription-coupled nucleotide excision repair plays an important role in the repair of polyG motifs in lung cancer, for which they found strong transcriptional bias for creating overlapping indels. The manuscript is written in a clear and an engaging manner.

What is disappointing, though, is that even though the work presented is entirely computational, with few exceptions the analysis is not performed according to good computational standards. Namely, instead of using statistical tests and p-values to assess enrichments or significance of the results, a reader is asked to “visually appreciate” the presented data. The data is also often presented as ratios, which can accumulate much higher error rates than individual numbers used for their calculations. These shortcomings are not allowing a reader to assess which of the presented results are truly significant and whether all conclusions drawn by the authors are supported by the data they present.

We thank the reviewer for an overall positive assessment and we seek to reassure the reviewer why particular methods were used to explain the results, or to visualise contrasting results, these were always assessed statistically. We have however,

incorporated additional references to the statistical methods used throughout the manuscript.

Specific remarks:

Point 3.1

Figure 1, all panels (a,b,c) would benefit from discussion of statistical significance of difference in discussed distributions. Comparing all possible pairs is not feasible due to the high number of different cancer types considered, but authors could e.g. compare distribution for each cancer with the total distribution for all cancer types considered or do pairwise comparisons for cancer types adjacent to each other in the figures.

We thank the reviewer for the comment. We compared the ratio of insertions to deletions across all patients to that for patients in each individual cancer type as the reviewer suggested. We find significant differences for multiple cancer types and calculated the associated z-scores resulting in the generation of a new supplementary figure (Supplementary Figure 1c) and performed Mann Whitney U tests comparing individual cancers to the total distribution for all cancer types, finding statistically significant differences for multiple cancer types. In addition, we added further statistical analyses for the panels in Figure 1, including Mann-Whitney U tests for figure 1a comparing the frequency of insertions and deletions and Levene's test for figure 1b comparing the variance of the size of insertions and deletions, from which we deduced that deletions displayed increased variance.

Point 3.2

Figure 2A. Current visualization does not give any information if the observed differences are significant. If distribution of ratios is close to normal, Z-transform will be helpful, i.e. expressing ratios in terms of their distance from the mean, expressed in standard deviations. For example, it will be more helpful to know that deep blue denotes 5 standard deviations above the mean than 1.4 mean, since the latter result may or may not be significant, while the former would be, even after applying correction for multiple hypothesis testing, such as the Bonferroni correction (here it will amount to multiplying p-values calculated for individual bins by the total number of bins). The reason multiple hypothesis correction (e.g. Bonferroni) should be applied is the authors consider 448 bins, so we expect to see 22 events (bins) with individually calculated p-value $= < 0.05$ (i.e. 1 in 20), which corresponds to the distance of 1.96 standard deviations from the mean. On the other hand, we will randomly observe values 5 sigmas away from the mean only once in 1,744,278 times, so such values will be clearly significant, even after correction for multiple hypothesis testing.

We thank the reviewer for the comment. We tested if the polyN motif densities are normally distributed using Kolmogorov-Smirnov and Shapiro tests and we found them to not follow a normal distribution. However, we provide an alternative non-parametric calculation of significance. We performed Mann-Whitney U tests with Bonferroni corrections, correcting for (number of bins tested) * (number of motifs tested), and generated a supplementary heatmap displaying the associated p-value in each bin

(Supplementary Figure 3a). The Mann Whitney U tests compared the density of polyN motifs in a bin versus across the bins and complement the enrichment scores displayed in Figure 2a. In addition, we performed the same type of statistical analysis for the supplementary figure 3c and generated a heatmap of p-values using Mann Whitney U tests with Bonferroni correction comparing the density of polyN motifs in the template and non-template strands (Supplementary Figure 3e).

Point 3.3

Figure 2C, D. Error bars need to be added, it is also not stated in which intervals density is computed

We thank the reviewer for the comment. We have added the error bars accordingly. We have also added the interval used (100bp) in the methods section.

Point 3.4

Figure 3D. Error bars need to be added.

We thank the reviewer for the comment. We have added the error bars accordingly.

Point 3.5

Line 170. The test used to computing p-value needs to be stated.

We thank the reviewer for the comment. We have now updated the test used.

Point 3.6

Lines 188 -192. P-value of the result should be computed.

We thank the reviewer for the comment. We have now calculated the p-value.

Point 3.7

Lines 194-200. No information is given on what criteria were used for dividing genes into highly- and lowly-expressed. How many genes were in each group? Fig. 3D show “polyG motifs at genes” – does it also includes previously used 4 bins outside of gene body?

Our apologies, this has now been corrected.

We have modified this to three expression groups as raised by the reviewer in Point 3.10 below, in-keeping with the rest of the paper. The separation of genes into the three groups (low, medium, high) was performed using the 33rd and 66th percentiles of RPKM levels across all genes for each cell of origin. Therefore, equal number of genes are found in each group. We have updated the Methods section to provide the information regarding the grouping of the genes.

This analysis does not include the footprint outside the gene body. It includes the regions between the transcription start and transcription end of each gene.

Point 3.8

Line 232 "specific stRands", not "stands"

Thank you, we have corrected this.

Point 3.9

Fig 4 B, C Error bars are necessary to assess results

Thank you, we have corrected this. We have also provided the statistics for both figures.

Point 3.10

Fig. 4 B and C shows the dependence of transcriptional strand asymmetry in indels for all cancer types. I understand before expression levels were analyzed for each cancer cell type separately, using cell-of-origin. How the genes were divided here into these three groups globally based and their varying expression levels in different conditions? Why two groups used before (high and low) were not used?

The genes in all cases were divided into equal groups using percentiles. However, to be consistent throughout the manuscript we have substituted the two groups in Fig.3D with three groups. The three groups represent expression levels of protein coding genes (relevant to the cell-of-origin of each cancer type), separated into the 33rd and 66th percentiles.

Point 3.11

The same consideration (need for error bars and statistical assessment of enrichments) apply to some supplementary figures.

We have added error bars to the supplementary figures (where relevant e.g. Supp Fig 6A-D) and further provided associated statistics of significance. We have calculated the significance of asymmetries at dinucleotide motifs across cancers (S10) as well as the MSI / MSS strand asymmetries at dinucleotides (S9). In addition, we have further our statistical analyses regarding the size of polyNs and strand asymmetries (S4) and added a supplementary figure controlling for the effect of replication direction (S5) and performed the associated statistics.

If these problems are satisfactorily addressed, I think the manuscript will be suitable for publication in Nature Communications.

We thank the reviewer for his comments and we have they have been satisfactory addressed. We believe the points raised have helped substantially improve the manuscript.

REVIEWERS' COMMENTS:

Reviewer #1 (Remarks to the Author):

Overall, most of the issues that were raised were addressed and I think the manuscript has been improved. Specifically, the authors excluded the replication bias as a confounder, did the effort to include NER-deficient samples in the analyses (although the analyses are not at all conclusive and I have some important concerns, please see below), improved the presentation of the figures and made the code available.

Major comments:

Point 1.3

The authors used a NER-defective sample to prove the role of TC-NER in the transcriptional strand asymmetry observed in tumors. They reason that if NER is not working, the asymmetry should not be observed. However, the mutated gene in the sample is XPC, which has an important role in GG-NER, but not a relevant role in TC-NER according to literature(1,2). Thus, the lack of transcriptional strand asymmetry observed in the sequenced sample cannot be attributed to TC-NER, given it should still be functional. This contradictory result does not go in the direction of TC-NER causing the asymmetry, and the authors should clarify it.

* For the authors' interest, more XPC-/- tumor samples (raw sequencing data) can be found in (2)

Marteijn et al, Understanding nucleotide excision repair and its roles in cancer and ageing, *Nat Rev Mol Cell Biol.* 2014 Jul; 15(7):465-81. doi: 10.1038/nrm3822.

Zhen et al, 2014, Transcription restores DNA repair to heterochromatin, determining regional mutation rates in cancer genomes, *Cell Rep.* 2014 Nov 20; 9(4):1228-34. doi: 10.1016/j.celrep.2014.10.031

Minor comments:

Point 1.9

I totally understand the authors' point. Please note my suggestion (it was not a requirement at all) was to use "similar" channels, specifically on having transcribed and not transcribed channels, to further explore which samples and which mutational processes present transcription asymmetry. However, I agree that having meaningful channels in the indels extraction is not trivial and I understand the authors might feel that the analysis is out of the scope of the manuscript.

Having asked to comment on concerns previously raised by Reviewer 2 and how these were addressed, I am attaching my comments.

Overall the authors have addressed the original concerns - I still have some comments though.

1. Neither the data nor the computer code used to analyse the data are accessible. This makes it impossible to replicate the results, or to validate whether there are issues in the analysis. I feel this is a non-starter for a computational biology paper. Specifically, I would like to see
 - URL(s) or accession number(s) for the mutation data
 - Uploads of code and other necessary files for the pre-processing and the statistical analysis on FigShare, Zenodo or similar
 - A bed file or similar with the exact indel coordinates and definitions used.

The code is available on Github and I had no problem to download it. I found some problems with tab indentations that made some scripts crash (for example while running `get_poly.py`) and should be fixed, but I managed to run some properly. That being said, it is hard to reproduce the presented work without the same input files. While I agree with the authors that they cannot share the mutations directly, they should share the scripts to download PCAWG data and do all the intermediate processing to get the same set of indels that was used in the manuscript. Then all downstream analyses should yield the same values as the authors'.

2. It is unclear to me whether the definition of "gene" used here refers to protein coding genes, or also to pseudogenes, lncRNAs, tRNA etc. The methods say: "Gene annotation from Ensembl was followed (Aken et al. 2016)" - Could the authors clarify, reference the URL, and include the gene definition with the code?

I think the way the authors get genic coordinates is one of the many correct ways they can be retrieved. However, in order to ensure reproducibility, they should specify the Ensembl version they used in the methods.

3. It is unclear how the authors treated exons and introns. Were both exons and introns considered jointly as the "gene"? Is the insertion/deletion strand asymmetry restricted to exons/introns only?

I agree the asymmetry should be restricted to exons/introns. Nevertheless, do the authors find similar levels of asymmetry in exons compared to introns?

Reviewer #3 (Remarks to the Author):

The authors generally answered all my requests satisfactorily. The only minor suggestion remaining is to use more contrasting colors for significant and non-significant p-values in supplementary figures 3A and 3E, currently it is not completely clear with used color gradient.

Response to Reviewers' comments

REVIEWERS' COMMENTS:

Reviewer #1 (Remarks to the Author):

Overall, most of the issues that were raised were addressed and I think the manuscript has been improved. Specifically, the authors excluded the replication bias as a confounder, did the effort to include NER-deficient samples in the analyses (although the analyses are not at all conclusive and I have some important concerns, please see below), improved the presentation of the figures and made the code available.

Major comments:

Point 1.3

The authors used a NER-defective sample to prove the role of TC-NER in the transcriptional strand asymmetry observed in tumors. They reason that if NER is not working, the asymmetry should not be observed. However, the mutated gene in the sample is XPC, which has an important role in GG-NER, but not a relevant role in TC-NER according to literature(1,2). Thus, the lack of transcriptional strand asymmetry observed in the sequenced sample cannot be attributed to TC-NER, given it should still be functional. This contradictory result does not go in the direction of TC-NER causing the asymmetry, and the authors should clarify it.

* For the authors' interest, more XPC^{-/-} tumor samples (raw sequencing data) can be found in (2)

Marteijn et al, Understanding nucleotide excision repair and its roles in cancer and ageing, Nat Rev Mol Cell Biol. 2014 Jul;15(7):465-81. doi: 10.1038/nrm3822.

Zhen et al, 2014, Transcription restores DNA repair to heterochromatin, determining regional mutation rates in cancer genomes, Cell Rep. 2014 Nov 20;9(4):1228-34. doi: 10.1016/j.celrep.2014.10.031

We thank the reviewer for the comment. We agree with the reviewer that more work is required to further understand the role of different genes in GG-NER and TC-NER and how they contribute to mutagenesis in these very rare tumors.

The lack of transcriptional strand asymmetry of insertions for this sample remains an interesting observation that requires more work. It would require systematic and larger collections of gene KO datasets for more genes in both TC-NER and GG-NER pathways in order to gain a better understanding. That is beyond the scope of this manuscript.

We have thus adjusted the text in the manuscript to address this point.

Minor comments:

Point 1.9

I totally understand the authors' point. Please note my suggestion (it was not a requirement at all) was to use "similar" channels, specifically on having transcribed and not transcribed channels, to further explore which samples and which mutational processes present transcription asymmetry. However, I agree that having meaningful channels in the indels extraction is not trivial and I understand the authors might feel that the analysis is out of the scope of the manuscript.

We thank the reviewer for the constructive comments in addressing and improving our manuscript.

Having asked to comment on concerns previously raised by Reviewer 2 and how these were addressed, I am attaching my comments.

Overall the authors have addressed the original concerns - I still have some comments though.

1. Neither the data nor the computer code used to analyse the data are accessible. This makes it impossible to replicate the results, or to validate whether there are issues in the analysis. I feel this is a non-starter for a computational biology paper. Specifically, I would like to see

- URL(s) or accession number(s) for the mutation data
- Uploads of code and other necessary files for the pre-processing and the statistical analysis on FigShare, Zenodo or similar
- A bed file or similar with the exact indel coordinates and definitions used.

The code is available on Github and I had no problem to download it. I found some problems with tab indentations that made some scripts crash (for example while running `get_poly.py`) and should be fixed, but I managed to run some properly. That being said, it is hard to reproduce the presented work without the same input files. While I agree with the authors that they cannot share the mutations directly, they should share the scripts to download PCAWG data and do all the intermediate processing to get the same set of indels that was used in the manuscript. Then all downstream analyses should yield the same values as the authors'.

We agree and while this paper was under review, the Pan-cancer analysis of whole genomes consortium papers were finally published. For patients from the ICGC part of the PCAWG consortium the set of mutations can now be obtained by the ICGC portal (<https://dcc.icgc.org/>) in the link: "[https://dcc.icgc.org/releases/PCAWG/consensus_snv_indel](https://dcc.icgc.org/releases/PCAWG/consensus_snv_indel)" with file name "final_consensus_snv_indel_passonly_icgc.public.tgz"

Unzipping of the downloaded file produces two folders one of which is the indel folder with the processed VCF files for each patient. For the TCGA part of the PCAWG consortium access control is required to access the data under the same link.

2. It is unclear to me whether the definition of "gene" used here refers to protein coding genes, or also to pseudogenes, lncRNAs, tRNA etc. The methods say: "Gene annotation from Ensembl was followed (Aken et al. 2016)" - Could the authors clarify, reference the URL, and include the gene

definition with the code?

I think the way the authors get genic coordinates is one of the many correct ways they can be retrieved. However, in order to ensure reproducibility, they should specify the Ensembl version they used in the methods.

In the previous version of the manuscript we updated to “Gene annotation from Ensembl was followed (Aken et al. 2016) and genes were downloaded from Biomart using Gene start and Gene end to define genes and filtering by only including protein-coding genes.”

Therefore, to answer the question only protein-coding genes were selected. We also provide the gene list with the code and now we have updated the steps section in the code to include the steps used to derive the genes annotation file used.

The link to the Biomart page we used is:

<http://grch37.ensembl.org/biomart/martview/c1d06f3affb6260c0cd7147bb4c3b6a8>

We selected default parameters but in the section Gene type we selected protein_coding and to define the gene boundaries we used Gene start (bp) and Gene end (bp). We also selected the attributes Strand and Gene name. Finally, we clicked Results and exported the TSV file with the genes.

3. It is unclear how the authors treated exons and introns. Were both exons and introns considered jointly as the "gene"? Is the insertion/deletion strand asymmetry restricted to exons/introns only?

I agree the asymmetry should be restricted to exons/introns. Nevertheless, do the authors find similar levels of asymmetry in exons compared to introns?

Yes, in general when assessing parts of the genome for transcriptional strand asymmetry, both exons and introns are included.

With respect to the second point: We have now performed this analysis. We found that there were differences between exons and introns both for polyGs and for polyTs with introns displaying a larger strand asymmetry towards the non-template strand relative to exons across cancer types (Mann Whitney U tests p-value < 0.05 for polyT motifs and p-value<0.01 for polyG motifs). These differences are not large even though the p-value is significant and we are disinclined to include this in the manuscript.

Reviewer #3 (Remarks to the Author):

The authors generally answered all my requests satisfactorily. The only minor suggestion remaining is to use more contrasting colors for significant and non-significant p-values in supplementary figures 3A and 3E, currently it is not completely clear with used color gradient.

We have now updated the two supplementary figures, changing the p-value to $-\log(p\text{-value})$, the coloring scheme and also adding the annotation *******, ******, ***** to represent p-values < 0.001, 0.001 and 0.05 respectively.